# The Nature and Quality of Australian Supermarkets’ Policies That Can Impact Public Health Nutrition, and Evidence of Their Practical Application: A Cross-Sectional Study

**DOI:** 10.3390/nu11040853

**Published:** 2019-04-15

**Authors:** Claire Elizabeth Pulker, Georgina S. A. Trapp, Jane Anne Scott, Christina Mary Pollard

**Affiliations:** 1School of Public Health, Curtin University, Kent Street, GPO Box U1987, Perth 6845, Australia; jane.scott@curtin.edu.au (J.A.S.); c.pollard@curtin.edu.au (C.M.P.); 2Telethon Kids Institute, The University of Western Australia, PO Box 855, West Perth 6872, Australia; gina.trapp@telethonkids.org.au; 3School of Population and Global Health, The University of Western Australia, 35 Stirling Highway, Crawley 6009, Australia

**Keywords:** supermarket, supermarket own brand, corporate social responsibility, food and nutrition policy, environmental sustainability, public health nutrition

## Abstract

Improving population diets is a public health priority, and calls have been made for corporations such as supermarkets to contribute. Supermarkets hold a powerful position within the food system, and one source of power is supermarket own brand foods (SOBFs). Many of the world’s largest supermarkets have corporate social responsibility (CSR) policies that can impact public health, but little is known about their quality or practical application. This study examines the nature and quality of Australian supermarkets’ CSR policies that can impact public health nutrition, and provides evidence of practical applications for SOBFs. A content analysis of CSR policies was conducted. Evidence of supermarkets putting CSR policies into practice was derived from observational audits of 3940 SOBFs in three large exemplar supermarkets (Coles, Woolworths, IGA) in Perth, Western Australia (WA). All supermarkets had some CSR policies that could impact public health nutrition; however, over half related to sustainability, and many lacked specificity. All supermarkets sold some nutritious SOBFs, using marketing techniques that made them visible. Findings suggest Australian supermarket CSR policies are not likely to adequately contribute to improving population diets or sustainability of food systems. Setting robust and meaningful targets, and improving transparency and specificity of CSR policies, would improve the nature and quality of supermarket CSR policies and increase the likelihood of a public health benefit.

## 1. Introduction

Poor diet is one of the most important risk factors for early deaths globally [1], and improving population diets is a public health priority [2,3]. The impact of corporations’ actions on public health has been described as the ‘corporate determinants of health’, recognizing that their influence can be positive or negative [4]. Transnational food producers have been identified as drivers of global obesity due to their supply of cheap, tasty and convenient foods that are persuasively marketed [5] with the primary objective of generating profit [6]. In addition, some food corporations have used their power to set policy agendas and influence government decisions that negatively impact public health [7], which is referred to as corporate political activity [8]. Globally, there has been a rapid growth in the proportion of foods sold from supermarkets, impacting population diets by making nutrient-poor processed foods more widely available [9,10,11]. In contrast, corporations that create jobs, pay their share of taxes, value and empower employees, including paying a living wage, and contribute to society can have a positive influence [4]. Therefore, holding corporations to account for actions that can impact public health is important [12]. 

### 1.1. The Impact of Supermarkets on Population Diets

Some of the ways that corporations can impact dietary intake are mapped in an ecological framework, which includes physical environments (e.g., supermarkets) and macro-level environments (e.g., food production and distribution systems) [13]. Physical environments or settings (e.g., supermarkets), also known as food environments, can support or undermine healthy eating [14]. The within-store food environment attributes of product, price, placement, promotion and provision of nutritional information can influence consumers’ food choice [15]. Supermarkets decide which products are available, how they are arranged on shelves, their price and promotions, which set boundaries on the choices available to consumers [16]. A number of practices that could negatively impact dietary intake have been identified in Australian supermarkets: less than half of supermarket packaged foods are classified as healthy [17]; snack foods, including crisps, chocolate and confectionery, are prominently displayed at supermarket ends-of-aisles and checkouts [18,19]; and foods designed to appeal to children are widely available and displayed in prominent supermarket locations [20]. 

Supermarkets are part of a highly complex global food system (i.e., the people and activities required to make food available) [21], often involving long supply chains [22]. The globalized food system significantly impacts population diets [23], influences environmental sustainability [24] and social justice [25] and has been described as invisible to consumers [26]. This is because globalization distances consumers from their food, with a lack of transparency over social, environmental and ethical decisions [25]. After being long overlooked, the environmental sustainability of food systems is now a priority issue for public health nutrition researchers [27]. 

The dominant neoliberal political context minimizes government regulation to promote global trade [28], so the ability of the global food system to support healthy and sustainable population diets is influenced by supermarkets that wield enormous power and influence [9], as well as transnational food producers [29]. In Australia, two supermarket chains dominate food retailing, accounting for 70% of grocery sales [30]. Supermarkets act as primary gatekeepers to the Australian food system, having gained power from many sources that overlap and reinforce each other [31]. Sources of power include instrumental (i.e., the ability to directly influence the decisions of other actors), structural (i.e., the ability to set limits on the choices available to other actors) and discursive (i.e., the use of communication practices to influence norms and values) [31]. Supermarket concentration has taken place in other developed countries including Austria, Canada, Denmark, Germany, France, Spain and the UK [32], which could indicate similar levels of supermarket power [31]. 

### 1.2. The Impact of Supermarket Own Brand Foods on Population Diets

Development of supermarket own brand foods (SOBFs) is a source of structural supermarket power [31]. Also known as private label, in-house brand, store brand, retailer brand or home brand [33], they are widely available in Australia and around the world [34]. Spain, the UK and Switzerland have the highest proportion of grocery sales from SOBFs (up to 45%) [35]. SOBFs provide a number of benefits to supermarkets, including increased control over the food system for greater returns [36], access to competitor information [37], increased leverage in negotiations with suppliers [38], higher profit margins [39] and flexible sourcing, with less dependence on local suppliers [40]. Australian SOBFs have impacted public health in many ways [31], including improving food safety standards [41], providing more affordable options [42], and increasing accessibility to affordable foods in the neighborhoods with supermarkets [36]. However, supermarkets determine what food is sold, which influences what food is produced [9]. Supermarkets have been instrumental in increasing availability of standardized cheap processed foods [40], and encouraging consumption of nutrient-poor foods [36]. They have shaped norms and values around foods to meet consumer needs [43], driving development of convenient SOBFs such as ready meals [44]. 

Few studies have described the contribution of SOBFs to population diets, comparing them to branded foods for sodium [45], other nutrients [46], serve size [46] and cost [47]. There were no consistent differences in nutritional quality across all foods, but some differences at the level of food category [45,46], and some cost savings [47,48]. Studies of the nutritional quality of SOBFs conducted in the Netherlands [49], the UK [50], Spain [51] and Ireland [52] found similarly inconsistent results. SOBFs in the Netherlands [49] and France [53] were significantly cheaper than the branded equivalent. There are no studies of the impact of SOBFs on any aspect of sustainability to the authors’ knowledge.

### 1.3. Policy Role of Supermarkets in Addressing Poor Diets and Promoting Sustainable Food Systems

Food policy action to encourage and support healthy dietary behavior has been mapped in a framework by Hawkes et al. (2013), which recognizes the influence of food environments and food systems, as well as individual-level behavior over population diets [23]. Policy levers can be ‘soft’ (e.g., voluntary initiatives) or ‘hard’ (e.g., price incentives, taxes, or regulations), and food corporations can contribute to improving population diets [54]. 

The United Nations (UN) and World Health Organization (WHO) have identified contributions profit-making corporations can make to addressing poor diets and promoting sustainable food systems. The 1987 UN Commission on Environment and Development proposed a global agenda for change to address the environmental impact of ongoing development, which called on corporations to accept more responsibility [55]. The 2004 WHO global strategy on diet, physical activity and health included a significant role for corporations, promoting healthy diets and making affordable, healthy foods widely available for consumers [56]. The UN Decade of Action on Nutrition resolution recommended corporations support governments to implement policies that address the burden of diet-related noncommunicable diseases whilst managing conflicts of interest [3]. 

Another influence over supermarket policymaking is the requirement for large companies operating within the European Union to disclose non-financial information relating to the environment, society and governance [57]. Companies are encouraged to provide forward-looking benchmarks and commitments, and unbiased reports that balance positive and negative aspects [57]. This focus on environmental impact but not public health is consistent with the focus of other global initiatives such as the Global Reporting Initiative [58], FTSE4Good [59], the Dow Jones Sustainability index [60] and the UN Global Compact [61].

### 1.4. Supermarket Voluntary Policies that can Impact Population Diets

Many of the world’s largest supermarkets have voluntary policies that impact population diets, referred to as corporate social responsibility (CSR) [62]. CSR includes voluntary policies of individual corporations, voluntary participation in public-private-partnership initiatives (e.g., Healthy Food Partnership in Australia [63]), or voluntary participation in industry-led initiatives (e.g., Australian Food and Grocery Council’s responsible children’s marketing initiative [64]). 

There are numerous CSR definitions, including political CSR theories which state that large, powerful companies need to act as good corporate citizens, taking responsibility for impacting society to protect their position and power [65]. Other CSR theories include: instrumental (i.e., corporations implement CSR as a means to generate profits), ethical (i.e., corporations implement CSR due to ethical concerns) and integrative (i.e., corporations implement CSR because continued support of society is essential for ongoing growth) [65]. 

Criticisms of CSR include being used as a means for food companies to prevent regulation [66], placing responsibility for selecting healthy foods onto consumers [67], allowing supermarkets to push problems such as food waste onto other parts of the food system [68] and that CSR documents have lacked details such as specific measures and independent assessment [69]. 

### 1.5. Ability of Supermarket CSR Policies to Address Poor Diets and Promote Sustainable Food Systems

Little is known about supermarket CSR policies that can impact population diets or sustainability. Previous research has shown that supermarkets were less active than food manufacturing and food service companies in having CSR policies to assist customers select nutritious foods in Australia [70]. UK supermarkets used CSR as a tool for competition [71], and could do more to support their customers to eat healthily [72]; US supermarkets tended to focus CSR efforts on social initiatives such as sponsoring local charities [73]; UK supermarkets’ CSR policies to encourage sustainable diets were assessed as weak, due to the focus on continuing financial growth [74]; and Swedish supermarkets were willing to support customers to make more sustainable food choices, provided they did not impact profits [75]. None of these studies assessed supermarket CSR policies relating to all aspects of public health nutrition (i.e., provision of safe, nutritious, affordable, secure and environmentally sustainable foods [76]), which include the attributes of accessibility, availability, cost and affordability, food preferences and choices, food safety and quality, nutritional quality and environmental sustainability [31], which represent an important gap in knowledge.

In addition to identifying supermarket CSR policies that can impact public health nutrition, it is important to analyze their quality, and provide evidence of their practical application. This is because supermarkets may not adhere to CSR policies, which can be influenced by their quality [77]. The Access to Nutrition Index (ATNI) monitors the contribution of the world’s largest food manufacturers to global nutrition issues by assessing the nutritional quality of their products and CSR policies [78]. An assessment of Australian supermarket CSR policies that can impact obesity prevention, based on the ATNI methodology, rated the comprehensiveness, specificity and transparency of CSR policies [79]. It concluded that Australian supermarkets needed to place more importance on nutrition within their corporate strategies to improve the healthfulness of supermarket environments [80]. 

Analyzing supermarket CSR policies could stimulate change throughout the food system [81]. Therefore, investigating the nature and quality of Australian supermarket CSR policies that can impact public health nutrition, and identifying evidence of their practical application, could lead to positive change. This study aimed to identify Australian supermarkets’ public health nutrition-related CSR policies, assess their quality and identify evidence of supermarkets putting them into practice for SOBFs.

## 2. Materials and Methods 

### 2.1. Study Scope

The study aimed to address two research questions: (1) What is the nature and quality of Australian supermarket CSR policies to improve public health nutrition? (2) Is there evidence of Australian supermarkets putting public health nutrition-related CSR policies into practice within their stores? For the purpose of this study, public health nutrition is defined as the provision of safe, nutritious, affordable, secure and environmentally sustainable food [76], and includes the following attributes: accessibility, availability, cost and affordability, food preferences and choices, food safety and quality, nutritional quality and environmental sustainability [31]. CSR is defined as voluntary policies specific to the supermarket, as well as voluntary participation in public-private-partnership initiatives or industry-led initiatives. 

Evidence of putting public health nutrition-related CSR policies into practice was collected via observational audits of SOBFs in stores. SOBFs were selected because they play a pivotal role both as a source of supermarket power and for their impact on public health [31], and supermarkets control SOBFs so they have more capacity to make the changes required to support dietary change compared with branded foods. 

Data collected from publicly available supermarket CSR policies were used to guide the analysis. Data collected from supermarket audits of SOBFs provided evidence of the CSR policies in practice.

### 2.2. Data Collection of CSR Policies

Websites for the main supermarkets present in Australia were searched for information referring to either CSR or sustainability. Coles Supermarkets Australia Pty Ltd (Coles) and Woolworths Supermarkets (Woolworths) together account for 70 percent of grocery sales in Australia [30]. Independent Grocers of Australia supermarkets (IGA) contribute a low overall share of grocery sales nationally, but represent the largest number of stores (over 50 percent) in Western Australia (WA) [48], so they were also included. Discount retailer Aldi was excluded from this study due to the limited range of foods sold [82], and because it had only just entered the WA market at the time of the study [83]. The website of Australia’s largest wholesaler Metcash, which supplies most products to the IGA network of independent stores and is responsible for marketing IGA [84], was also searched for information referring to CSR or sustainability. CSR policies available on the websites were included as research materials.

### 2.3. Data Collection of CSR Evidence

#### 2.3.1. Selection of Supermarkets to Audit

In-store observational audits of all SOBFs were conducted in three purposely selected supermarkets, one for each chain. The stores were selected because they were ‘optimized’ (i.e., large supermarkets with an increased likelihood of displaying most of the available SOBFs, and with the most up-to-date layouts and displays) [85], and conveniently located in Perth, WA. The Woolworths ‘next generation’ store had been recently extensively refurbished [86]. The IGA was the WA ‘IGA store of the year’. The Coles store was located close to parent company Wesfarmers’ offices, meaning it would receive ongoing scrutiny from senior executives. The supermarket audit methods involved collecting observational data in a systematic way from each of the selected supermarkets. The protocol for the audits is provided in detail elsewhere and described briefly below [85].

#### 2.3.2. Identification of SOBFs

SOBFs were identified by the presence of supermarket branding on the front-of-pack. Online shopping websites for Coles and Woolworths were searched to generate lists of SOBF, which assisted with identifying the SOBFs during the audits [87,88]. All packaged foods and non-alcoholic beverages (referred to as food hereon in) carrying a supermarket own brand (SOB) were audited, including pre-packed fresh products such as fruits, vegetables and meat that carried the supermarket’s name on the label. A list of SOBs is provided in Appendix A. 

#### 2.3.3. Data Collection from Supermarkets

Two researchers visited each supermarket together to conduct the audits during a three-week period in February 2017. This timing avoided product changes that occur during Christmas, the Australian school summer holiday period, and prior to Easter. All aspects of supermarket food environments were audited including products available, price, placement, promotion and provision of nutrition information [15]. Photographic images were taken of the front-of-pack, shelf-edge label, location of the product and promotions for all SOBFs present during the audit period and filed electronically. Back-of-pack information, which typically includes the ingredients list and nutrition information panel, was not collected in store, as the intent was to reflect a typical consumer shopping experience where purchase decisions are made within a few seconds [89], indicating that little time is spent consulting the back-of-pack information.

Data were extracted from the photographs into a database that was constructed in Microsoft Excel (Version 2013, Redmond, Washington, USA). Pre-coded responses were established for each column of data for consistency of the data entry. Product groups were assigned based on supermarket layouts, where similar products were displayed together. Within each product group (e.g., bakery and desserts) food groups were also assigned (e.g., bread). 

Data entry for the first product group was piloted to ensure all relevant information from the photographs was captured, and to establish the suitability of the pre-coded responses. Both researchers who collected the data completed the data entry, which was then reviewed for accuracy and changes made by the first author as required to ensure consistency.

### 2.4. Assessment of CSR Policies

Content analysis of the CSR research materials was conducted by applying a framework of supermarket impacts on public health [62], focusing on policies that related to the public health nutrition attributes of accessibility, availability, cost and affordability, food preferences and choices, food safety and quality, nutritional quality and sustainability [31]. 

A political CSR lens (i.e., whereby large, powerful companies need to act as good corporate citizens, taking responsibility for impacting society to protect their position and power [65]) guided the analysis of supermarket public health nutrition-related CSR policies. Political CSR theories refer to the power held by large companies, which demands that they act responsibly [90] as good corporate citizens [65] in order to protect their power and position. Any reference to public health nutrition attributes in the CSR policies was recorded and summarized in a matrix constructed in Microsoft Excel (Version 2013, Redmond, Washington, USA). The quality of CSR policies was classified as ‘clear and specific’, or ‘vague or not specific’, to show the variability in the types of policy statements made.

### 2.5. Assessment of Store Audit Data

#### 2.5.1. Nutritional Quality

The nutritional quality of SOBFs was assessed using front-of-pack information only (i.e., product name, product description, and the Health Star Rating (HSR) nutrition label). Foods were categorized as: (i) nutritious (i.e., from the recommended five food groups), or (ii) nutrient-poor (i.e., ‘discretionary’ foods, which should only be eaten sometimes, and only in small amounts) based on the Australian Guide to Healthy Eating (AGTHE) [91]. The Educator’s Guide [92] and the Australian Bureau of Statistics’ principles for identifying discretionary foods [93] guided the assessment.

#### 2.5.2. Food Preferences and Choices 

Presence of any statement, claim or logo that related to any aspect of nutrition, health or sustainability potentially influencing food selection was recorded. The classification of nutrition and health statements and claims was guided by a taxonomy that identified nutritional information, nutrition claims, health claims and marketing statements and claims [94], including the government-led HSR, which aims to assist consumers to select healthier foods [95]. Sustainability statements and claims were grouped as animal welfare, food and packaging waste or sustainable sourcing (e.g., Fairtrade coffee). 

#### 2.5.3. Other Supermarket Audit Data

Other front-of-package statements and claims recorded related to whether SOBFs were made without specific ingredients such as allergens, or suitable for specific dietary preferences (e.g., vegetarian). 

Data extracted into the database also included the shelf location of each SOBF, the techniques used to make the SOBFs prominent (e.g., placing the SOBFs adjacent to its branded equivalent), the presence of price promotions and the presence of messages indicating value for money.

## 3. Results

### 3.1. Australian Supermarket CSR Policies

Fifty-one CSR policies that can impact public health nutrition were made by Australian supermarkets, summarized in Table 1. There were more CSR policies made by Coles (51%) and Woolworths (41%) than IGA (8%) (Table 2). Over half (61%) of supermarket CSR policies related to an aspect of sustainability (i.e., animal welfare, including sustainable fishing, food and packaging waste and product and ingredient sourcing). Few of the CSR policies related to accessibility (2%) or affordability (4%), and none to availability; none of the policies that were present related to SOBFs. Some CSR policies described the importance of ensuring SOBFs are nutritious (18%) or safe (8%).

Half of the supermarket CSR policies were clear and specific (Table 2). The vague or not specific CSR policies referred to nutrient reduction, the amount of food waste sent to landfill, food safety standards and affordability initiatives, but did not provide targets or details of current practices. Fifty-eight percent of CSR policies were clear and specific for sustainability.

### 3.2. Practical Application of CSR Policies 

#### 3.2.1. Accessibility, Availability, and Affordability

Audit data showed the extent to which the supermarkets made nutritious SOBFs available, accessible and affordable. Availability of nutritious SOBFs varied between supermarkets, with Woolworths making the largest proportion available (54%) (Table 3). Eleven percent of nutritious SOBFs were made accessible by the supermarkets by locating them on the most prominent shelf. In Coles, 71% of available nutritious SOBFs were highlighted with a pricing message on the shelf-edge (i.e., signaling every day value, not a special discounted price), which was also present for 26% of IGA and 5% of Woolworths SOBFs. Seventy-five percent of nutritious SOBFs across all supermarkets were prominently placed adjacent to their branded equivalent, or co-located with a range of SOBFs in a block. Six percent of SOBFs were price-promoted (i.e., displayed a special discounted price).

#### 3.2.2. Food Preferences and Choices

CSR policies made by Coles and Woolworths to apply the HSR to all SOBFs were not achieved in practice. HSR was only present on 66% of Coles and 51% of Woolworths SOBFs (Table 2). Nutrition and health-related statements and claims implying foods are nutritious choices were present on 66% of SOBFs. Health marketing techniques, including emphasis on naturalness and promotion of balance or goodness, were common on SOBFs in Woolworths (73%) and Coles (69%), but not IGA (18%). Nutrition claims were also used on a larger proportion of Woolworths (25%) and Coles (20%) SOBFs compared to IGA (12%). Health claims, including health endorsements, were present on few (2%) SOBFs in any supermarket. Eleven percent of SOBFs suitable for special dietary requirements were available in Coles, consistent with their CSR policy, which did not provide any specific targets.

#### 3.2.3. Food Safety and Quality

No statements about compliance with food safety standards were made on the front-of-pack of any SOBF. All supermarkets had CSR policies related to the avoidance of artificial colors, flavors, MSG or genetically modified ingredients (Table 1). Labeling claims about the absence of these artificial ingredients were present on 61% of Coles and 60% of Woolworths SOBFs (Table 2). 

#### 3.2.4. Nutritional Quality

Coles and Woolworths set targets for nutrient reduction which could not be assessed because the targets were not specified (Table 1). The CSR commitment by Woolworths for new SOBFs to improve the nutritional quality of their product portfolio was also not specific enough to enable verification.

#### 3.2.5. Sustainability: Animal Welfare

Cage-free eggs were committed to by Coles and Woolworths; however, audits found not all were free-range (Coles: 3 of 4; Woolworths: 1 of 3; IGA: 0 of 3) (Table 4). Coles and Woolworths had CSR policies relating to sourcing fish and seafood certified as sustainable (Table 1). Almost all SOBF fish products (i.e., frozen fish, canned fish and packaged fresh fish) made statements or claims about sustainable fishing, with some products making more than one claim (Table 4). Coles had CSR policies to protect animal welfare for beef, chicken and pork, with statements and claims present on 48% of all Coles SOB meat products (e.g., bacon, burgers, canned meat and packaged fresh meat). Woolworths made CSR policies to protect animal welfare for chicken, and chicken carried the most animal welfare statements and claims present (59%). 

#### 3.2.6. Sustainability: Product and Ingredient Sourcing

All supermarkets had CSR policies to source certified sustainable palm oil, but there were no statements or claims made on the front-of-pack of SOBFs. It is likely that statements would be made in the ingredients list on the back-of-pack. Other CSR policies from Coles and Woolworths described efforts to source ethically certified coffee, tea, cocoa and chocolate or sugar (Table 1). Ethical sourcing statements and claims were present on 59% of specified SOBFs in Coles, 56% of specified SOBFs in Woolworths and 1% of specified SOBFs in IGA (Table 5). They were most prevalent on sugar (80% of SOBF sugar products). Ethical sourcing certification logos present included Fairtrade, the Fairtrade cocoa program, Rainforest Alliance, Rainforest Alliance cocoa, UTZ and Bonsucro.

## 4. Discussion

This unique study analyzed the presence and quality of Australian supermarkets’ CSR policies related to nine attributes of public health nutrition, and identified evidence of supermarkets putting them into practice. This is important because practical application of CSR has a more direct influence on food environments than CSR policies [106].

### 4.1. Implications of CSR Policies

Coles and Woolworths had more CSR policies that can impact public health nutrition compared with Metcash, suggesting a stronger commitment. This finding is consistent with the political CSR lens applied, whereby the most powerful Australian supermarkets [31] had more CSR policies, which may assist in protecting their position and power [90]. Wesfarmers (which owns Coles) and Woolworths make some important contributions to society: they are Australia’s two largest companies by revenue [107], employing 418,000 people [97,100]. However, the supermarkets are unlikely to contribute to positive dietary change without CSR policies addressing the fundamental issues of accessibility, availability and affordability of nutritious SOBFs [76]. 

Coles, Woolworths and Metcash are members of the government-led Healthy Food Partnership, which aims to improve the health of all Australians; however, the initiative is still in development after three years [108]. Coles and Woolworths have also been key supporters of the HSR nutrition label launched in 2014 [95], but their uptake of the scheme has been slow, and a number of issues including a lack of transparency have been identified [109]. The lack of supermarket CSR policies that can contribute to positive population dietary change is a weakness in the current policy approach, particularly in the context of limited Australian government policy action to improve population diets since 2010 [110]. 

### 4.2. Quality of CSR Policies

Australian supermarket CSR policies varied in quality: half lacked specificity, providing vague or ambiguous statements that could not be assessed. A UK study of the quality of supermarket CSR policies to remove unhealthy food from checkouts found that supermarkets that provided specific details had good levels of adherence [77]. Voluntary initiatives including CSR maintain credibility by being transparent, and specifying benchmarks or targets to enable objective evaluation [111]. Australian supermarkets can gain credibility for their CSR efforts to impact public health nutrition by providing specific details, setting transparent targets and regularly reporting progress made. 

### 4.3. Evidence of Putting CSR Policies into Practice

Despite the absence of CSR policies relating to the availability of nutritious SOBFs, this study revealed that all supermarkets had nutritious SOBFs available, but the proportion varied considerably between supermarkets. This could be addressed with CSR policies that set targets for the proportion of SOBFs that are nutritious, using transparent criteria to assess nutritional quality. Supermarkets Ahold Delhaize [112] and Marks and Spencer [113] have such targets in place, aiming to increase the proportion of sales from nutritious SOBFs. 

Audit findings also indicated that supermarkets could do more to make nutritious foods accessible, by placing them on shelves at eye-level, offering price promotions and using messages on shelf-edge labels. The shelf-edge labeling system Guiding Stars, used in some US and Canadian supermarkets, has been applied to all products to guide consumer selection of nutritious foods [114]. Australian supermarkets could apply the HSR to shelf-edge labels to guide consumer food selection across all food, not just those that are packaged. CSR policies to improve the accessibility of nutritious foods in Australian supermarkets are needed as a priority. 

Australian supermarkets should also consider implementing CSR policies to ensure that nutritious SOBFs cost no more than nutrient-poor SOBFs, following the example set by Tesco [115]. Government-led market basket surveys which monitor the cost of healthy foods suggest that SOBFs are cheaper than the branded equivalents [48]. The Woolworths initiative to develop an affordable healthy eating index indicates potential public health benefit. However, the index is a subjective measure of consumer opinion [116], not an objective measure of the affordability of healthy foods at Woolworths. For public health impact, an objectively derived index is recommended, with transparency over the foods included and criteria used to define ‘healthy’ and ‘affordable’. 

Coles CSR policies stated they set product safety and quality standards, which they require suppliers to meet. They referred to audits of suppliers, and disclosed the number of products recalled due to product safety issues [97]. Yet their CSR policies were vague and did not provide any specific details or targets. No other reference to the importance of food safety was provided by any supermarket. These findings are consistent with previous research, which found that supermarkets enforce rules about acceptable food safety and product quality to manage reputational risk [117], and their standards are typically more stringent than government food safety standards [118]. Suppliers are required to provide assurance of food safety to enable them to do business with supermarkets [119]; however, these standards are not communicated to consumers on labels [120]. Supermarkets make important decisions regarding food safety risks that affect public health [121], suggesting that increased transparency regarding the standards set and levels of compliance is needed. 

Supermarkets influence population health by determining the contribution of added fats, sugars and salt to SOBFs [40]. Targets to reduce sodium, sugar and saturated fat in SOBFs were referred to by Coles and Woolworths, but details of the targets were not provided. The Woolworths CSR policy that new SOBFs would improve the nutritional quality of the range also lacked detail about how this would be achieved. Making nutrient reduction targets publicly available is important [79] to increase transparency and credibility [111]. 

Nutrition and health statements and claims that imply foods are nutritious choices, were widely used on Coles and Woolworths SOBFs. CSR policies to ensure these statements and claims are only used on nutritious SOBFs that are consistent with the AGTHE five food group foods [91] are recommended to prevent deceptive marketing practices [94].

### 4.4. CSR Policies to Support Sustainability

Over half of supermarket CSR policies related to the sustainability attribute, including setting animal welfare and ethical sourcing standards. Coles and Woolworths committed to animal welfare standards for SOBFs in a number of ways, for example, only selling only cage-free eggs, sustainably sourced fish and seafood and ensuring the five freedoms of animals were upheld [122]. These animal welfare standards are an important step, but do not extend far enough to have a meaningful impact, as they have established a consumer-driven model of animal welfare, rather than enforcing the welfare of all farmed animals [123]. Not all relevant SOBFs were labeled with animal welfare certification, indicating CSR policies were not achieved. In addition, the use of labeling to highlight sustainability standards has been challenged by the assertion that all food should be sustainable [27]. This is important, because Australians reportedly lack the knowledge and motivation to select foods consistent with environmental sustainability [124], even though third-party certification of sustainability standards, such as ethical sourcing or animal welfare, guarantees adherence [125]. 

Whilst sustainable sourcing initiatives can contribute to improving some aspects of the food system, they do not address the bigger issue of encouraging healthy and sustainable population diets [126]. The supermarket CSR policies did not refer to healthy and sustainable diets, which may be because there is not one commonly agreed upon definition [75] or approach to describing the level of sustainability of a diet [127]. One definition of healthy and sustainable diets is: reducing overconsumption, reducing the amount of nutrient-poor discretionary foods eaten and replacing animal-based foods with plant-based foods [128]. As discretionary foods accounted for a third of Australian diet-related environmental impacts, a reduction in production and consumption would have a significant impact [129]. Australian supermarkets should introduce CSR policies to reduce the production and consumption of discretionary foods and other foods that have high environmental impacts, such as meat, to encourage healthy and sustainable diets in a more meaningful way. For example, two Swedish supermarket chains have campaigns to encourage consumers to reduce meat consumption and eat more vegetarian food instead [130]. 

### 4.5. Implications of Conducting Supermarket Audits 

Practical application of CSR (i.e., actions that have been implemented) has a more direct influence on food environments than CSR policies (i.e., a statement of intent), so including product information in monitoring is recommended [106]. Whilst the inclusion of supermarket audits to evaluate how well CSR policies were practically applied in stores was time consuming, this study revealed several advantages, including (i) identifying specific CSR policies that fill gaps with the potential for public health nutrition impact, (ii) exposing weaknesses in the practical application of supermarket CSR policies and (iii) assessing supermarket CSR policies and practical applications across all aspects of food environments that can influence consumer food selection. 

Identifying specific CSR policies with the potential for impact to fill existing gaps is a priority. For example, there were no effective CSR policies to improve the availability, accessibility or affordability of nutritious SOBFs. The supermarket audit data provided information about the supermarkets’ current performance, enabling recommendations for specific CSR policies with the potential for public health nutrition impact.

Analyzing supermarket audit data can expose weaknesses in the practical application of CSR policies. Coles and Woolworths had CSR policies to implement HSR on all SOBFs, yet the supermarket audits revealed the policies have not been achieved. Similarly, audit data revealed Coles and Woolworths’ animal welfare standards were only applied to specific SOBFs, and were therefore more limited in reach than CSR policies implied. These findings indicate supermarkets should be held accountable to an empowered independent body for fulfilling their CSR policies [7,131]. 

Finally, including supermarket audit data, rather than product information, means the impact of supermarket CSR policies on all aspects of food environments that can influence consumer food selection (i.e., the products available, their price, promotion, placement and provision of nutrition information [15]) can be assessed, and what can be measured is more likely to be acted on.

This study’s findings indicate that setting robust and meaningful targets, improving transparency and specificity of CSR policies, and regularly updating progress in CSR reports would improve the nature and quality of supermarket CSR policies to benefit public health. Measures to hold supermarkets accountable for fulfilling their CSR policies would assist in improving translation into practice. Researchers in other countries with high proportions of grocery sales from SOBFs (e.g., Spain, the UK and Switzerland) may find conducting a similar analysis assists in identifying the ability of supermarket CSR policies to contribute to improving population diets and the sustainability of food systems.

### 4.6. Strengths and Limitations

This is the first study to comprehensively analyze all of the attributes of public health nutrition that can be impacted by supermarket CSR policies, including environmental sustainability. The strengths of this study include the extensiveness of the in-store data collection from audits of three purposively selected supermarkets to describe the nature and extent of SOBFs. This comprehensive information provided evidence of supermarkets’ translation of public health nutrition-related CSR policies into practice. Quality of CSR policies was also reported. A number of limitations relate to this study. Back-of-pack information present on SOBFs was excluded, so nutrition information panels, ingredient lists and allergen declarations were not assessed as evidence of supermarkets putting CSR policies into practice. Some CSR policies may relate to this information, and therefore results for evidence of supermarkets putting CSR policies into practice may be understated. Only publicly available supermarket CSR policies were included in this study, so other work may be in progress that can positively influence public health nutrition. CSR policies that related to internal initiatives could not be verified in this study (e.g., working groups to deliver healthier SOBFs). Although this study focused on supermarket CSR policies that can impact public health nutrition, other food system actors, including government, food manufacturers, and food service operators, have important contributions to make. Further research to investigate policies that can impact public health nutrition from these actors is needed.

## 5. Conclusions

Corporations, including supermarkets, have been charged with contributing to improving population diets and the sustainability of food systems. This is particularly important for countries with limited government public health nutrition policy action, such as Australia. Supermarket CSR policies in Australia can impact public health nutrition, but few addressed accessibility, availability or affordability of nutritious SOBFs. All supermarkets sold nutritious SOBFs and used marketing techniques to make them visible in store. Sustainable sourcing CSR policies were only implemented for some SOBFs, and did not address the bigger issue of supporting healthy and sustainable diets. Half of the supermarket CSR policies lacked specificity, providing vague or ambiguous statements that could not be assessed. These findings suggest Australian supermarket CSR policies are not likely to adequately contribute to improving population diets or sustainability of food systems. Recommendations for supermarket CSR policies in Australia and other countries include: provide specific details, set transparent targets and report progress against these targets; set targets for the availability, accessibility and affordability of nutritious SOBFs; reinforce the importance of food safety and quality by making existing standards more transparent; specify nutrient reduction targets, and report progress made for SOBF reformulation; and ensure sustainable sourcing policies encompass all SOBFs, with a focus on supporting healthy and sustainable diets.

## Figures and Tables

**Table 1 nutrients-11-00853-t001:** Summary of Australian supermarket CSR policies to improve public health nutrition ^#^.

Public Health Nutrition Attribute	Supermarket CSR Policies					
	Coles		Woolworths		Metcash	
	Policies	Quality	Policies	Quality	Policies	Quality
Accessibility (i.e., location of stores, location of products, education and promotion initiatives to support selection of healthy foods)	None	N/A	Free fruit is available for any child shopping with an adult.	Clear and specific	None	N/A
Availability (i.e., availability of foods to meet specific needs, including healthy and sustainable)	None	N/A	None	N/A	None	N/A
Food cost and affordability (i.e., makes healthy foods affordable)	None	N/A	The Affordable Healthy Index will be developed to help customers choose healthier and affordable baskets of foods.	Vague or not specific	None	N/A
			Marketing campaigns where prices of healthier products are reduced, and tips and swaps for healthier eating are provided in stores.	Vague or not specific		
Food preferences and choices (i.e., assists consumers to select the foods that meet specific needs, and encourages selection of healthy or sustainable choices, via provision of information such as labels and signs)	The Health Star Rating (HSR) is applied to the front-of-pack of SOBFs (1633 products).	Clear and specific	The Health Star Rating is applied to the front-of-pack of all eligible SOBFs (2200 products).	Clear and specific	None	N/A
	A sourcing policy that prioritizes Australian-grown food is in place. Eighty percent of SOBFs are sourced in Australia.	Clear and specific				
	SOBFs suitable for customers with special dietary requirements are provided (e.g., gluten free, vegetarian).	Vague or not specific				
Food safety and quality (including traceability, hygienic stores and avoidance of specific ingredients that are perceived to be harmful)	Coles works with suppliers to ensure SOBFs are safe and high quality (e.g., by setting the Coles Manufacturing Supplier Standards).	Vague or not specific	SOBFs do not contain artificial colors or flavors, or MSG.	Clear and specific	SOB ‘Community Co’ foods excludes artificial flavors and colors and genetically modified ingredients.	Clear and specific
	SOBFs do not contain 28 artificial colors, and other additives are not used when possible.	Clear and specific				
Nutritional quality (including foods, nutrients and portion sizes that support healthy eating)	Targets to reduce sodium, sugar and saturated fat have been set for SOBFs. Prioritized product ranges include ‘nutritional snacks and cereals’ and sausages.	Vague or not specific	Targets to reduce sodium, sugar and saturated fat have been set for SOBFs.	Vague or not specific	None	N/A
	Coles has an internal working group, including nutritionists, which focused on delivering healthier choices across SOBFs.	Vague or not specific	New SOBFs will improve the nutritional quality of the product portfolio.	Vague or not specific		
			Woolworths has a cross-functional health working group, including nutritionists, and is supported by the executive committee to embed a health strategy.	Vague or not specific		
Sustainability: animal welfare (e.g., sustainable fishing practices, sells cage-free eggs, bans products due to animal welfare concerns)	SOB animal welfare standards are based on the five freedoms of animals.	Vague or not specific	Own brand eggs are cage free.	Clear and specific	Phasing out SOB cage eggs by the end of 2018.	Clear and specific
	SOB eggs are cage-free with animal welfare certification (e.g., RSPCA).	Clear and specific	Adopted an animal welfare standard for SOB Farmers Own milk.	Vague or not specific		
	SOB fish and seafood is certified by the Marine Stewardship Council, Aquaculture Stewardship Council or meets Coles Responsibly Sourced seafood criteria.	Vague or not specific	SOB fish and seafood is certified by the Marine Stewardship Council, Aquaculture Stewardship Council, Best Aquaculture Practice and Global GAP.	Clear and specific		
	SOB beef has no added hormones, and antibiotics are only allowed for animal health purposes under veterinary supervision.	Clear and specific	SOB seafood products are labeled with certification eco-labels.	Clear and specific		
	SOB chicken meat is from suppliers with animal welfare certification (e.g., RSPCA).	Clear and specific	All SOB fresh chicken is from suppliers with RSPCA certification.	Clear and specific		
	SOB pork is sow-stall free.	Clear and specific				
Sustainability: food and packaging waste (e.g., reduce food waste, source packaging materials from sustainable sources)	Target set to work with suppliers to halve food waste.	Vague or not specific	Targets set to reduce the amount of food sent to landfill by reducing stock loss, improving store waste management and working with farmers.	Vague or not specific	Aim to reduce waste sent to landfill.	Vague or not specific
	Reusable plastic crates have been introduced for fruit, vegetables, poultry, red meat and salads to reduce product damage and reduce food waste.	Vague or not specific	The SOB ‘Odd Bunch’ was created to sell misshapen fresh fruit and vegetables at affordable prices.	Vague or not specific		
	Launched three fresh produce SOBFs that use vegetables, which would otherwise contribute to landfill.	Vague or not specific	The packaging format for SOB potato and pasta salads was changed to reduce the amount of plastic used.	Vague or not specific		
	Launched a SOB banana bread, which uses bananas that would have otherwise gone to landfill.	Vague or not specific				
	SOB fresh beef, pork and lamb mince are packaged in an ultra-high barrier renewable and recyclable material.	Vague or not specific				
	SOB packaging will be recyclable by 2020.	Clear and specific				
Sustainability: product and ingredient sourcing (e.g., coffee, soy, organic)	Only Certified Sustainable Palm Oil is used in SOBF.	Clear and specific	Only Certified Sustainable Palm Oil is used in SOBF.	Clear and specific	Only Certified Sustainable Palm Oil is used in SOBFs.	Clear and specific
	Palm oil is specifically identified on ingredients lists rather than the generic term ‘blended vegetable oils’ in SOBFs.	Clear and specific	SOB sugar will be certified by Bonsucro.	Vague or not specific		
	SOB coffee is certified by UTZ, Fairtrade or Rainforest Alliance.	Clear and specific	A SOB range of sustainably certified tea will be launched in 2018, with all own brand tea certified by 2020.	Clear and specific		
	SOB tea is certified by UTZ, Fairtrade or Rainforest Alliance.	Clear and specific				
	SOB cocoa and chocolate will be from certified sources by 2020.	Clear and specific				

# Sources of supermarket CSR policies [96,97,98,99,100,101,102,103,104,105]; SOBF is supermarket own brand food, SOB is supermarket own brand, N/A is not applicable, RSPCA is Royal Society for Prevention of Cruelty to Animals.

**Table 2 nutrients-11-00853-t002:** Number of Australian supermarket CSR policies to promote public health nutrition^#^.

Public Health Nutrition Attribute	Supermarket CSR Commitments					
	Coles		Woolworths		Metcash	
	Policies	Clear and Specific	Policies	Clear and Specific	Policies	Clear and Specific
Accessibility	0	-	1	1	0	-
Availability	0	-	0	-	0	-
Food cost and affordability	0	-	2	0	0	-
Food preferences and choices	3	2	1	1	0	-
Food safety and quality	2	1	1	1	1	1
Nutritional quality	4	0	5	0	0	-
Sustainability: animal welfare	6	4	5	4	1	1
Sustainability: food and packaging waste	6	1	3	0	1	0
Sustainability: product and ingredient sourcing	5	5	3	2	1	1
Total	26	13	21	9	4	3

# Sources of supermarket CSR policies [96,97,98,99,100,101,102,103,104,105].

**Table 3 nutrients-11-00853-t003:** Evidence of Australian supermarkets putting public health nutrition-related CSR policies into practice.

Supermarket Audit Findings	Coles		Woolworths		IGA	
	Frequency	Percent	Frequency	Percent	Frequency	Percent
***Nutritious SOBFs as assessed using the principles of the AGTHE ^*^***						
Available nutritious SOBF	830	47.9%	969	53.5%	141	35.5%
Located on most prominent shelf	211	12.2%	190	10.5%	23	5.8%
Other prominence techniques used	737	42.6%	582	32.1%	136	34.3%
Price promotions present	36	2.1%	79	4.4%	6	1.5%
Everyday low-pricing message present	589	34.0%	45	2.5%	37	9.3%
Other pricing message present	38	2.2%	93	5.1%	24	6.0%
***Food preferences labeling statements and claims***						
No artificial colors/flavors/preservatives/MSG	1063	61.4%	1082	59.7%	29	7.3%
Allergen advice	144	8.3%	98	5.4%	32	8.1%
Certified organic/organic	48	2.8%	126	7.0%	0	0.0%
Vegetarian/vegan product	89	5.1%	25	1.4%	0	0.0%
GI claims	48	2.8%	0	0.0%	0	0.0%
***Front-of-pack nutrition labels***						
Health Star Rating present	1141	65.9%	923	50.9%	0	0.00%
Daily Intake Guide present	215	12.4%	465	25.7%	313	78.8%
***Nutrition and health statements and claims***						
Nutrition claims present	348	20.1%	450	24.8%	49	12.3%
Health claims present	14	0.8%	52	2.9%	11	2.8%
Health marketing techniques present	1198	69.2%	1316	72.6%	71	17.9%
***Sustainability statements and claims***						
Sustainable fishing statements and claims	65	3.8%	68	3.8%	27	6.8%
Animal welfare statements and claims	185	7.1%	81	4.5%	1	0.3%
Sustainable sourcing statements and claims	54	3.1%	37	2.0%	4	1.0%
**Total audited products**	**1731**		**1812**		**397**	

^*^ AGTHE is the Australian Guide to Health Eating [91]; MSG is monosodium glutamate; GI is gastrointestinal index; SOBF is supermarket own brand food, WA is Western Australia.

**Table 4 nutrients-11-00853-t004:** Sustainability statements and claims present on Australian supermarket own brand foods: animal welfare.

Sustainability Statement or Claim	Coles		Woolworths		IGA	
	Frequency	Percent	Frequency	Percent	Frequency	Percent
***Cage-free eggs***						
Free-range eggs	3	75.0%	1	33.3%	0	0.0%
*Total audited eggs*	*4*		*3*		*3*	
Made with free-range eggs	7	0.4%	7	0.4%	0	0.0%
***Sustainable fish***						
Responsibly caught/sourced/farmed fish	54	84.4%	13	20.3%	1	5.0%
Dolphin friendly/dolphin safe: drift net free	0	0.0%	31	48.4%	13	65.0%
Certified sustainable seafood MSC/Alaska Seafood logo	11	17.2%	11	17.2%	3	15.0%
Pole and line caught	0	0.0%	12	18.8%	1	5.0%
FAD free tuna	0	0.0%	0	0.0%	9	45.0%
Wild caught	0	0.0%	1	1.56%	0	0.0%
*Total audited fish and seafood*	*64*		*64*		*20*	
***Five freedoms of animals***						
RSPCA approved	55	28.2%	34	17.4%	0	0.0%
Sow stall free pork	43	22.1%	0	0.0%	1	3.5%
Free-range meat	9	4.6%	16	8.2%	0	0.0%
From hens free to naturally roam and perch	4	2.1%	0	0.0%	0	0.0%
Pasture fed/grass fed	0	0.0%	1	0.5%	0	0.0%
Outdoor bred	0	0.0%	1	0.5%	0	0.0%
***Use of hormones or antibiotics***						
Antibiotic free	0	0.0%	1	0.5%	0	0.0%
No added hormones beef	64	32.8%	20	10.2%	0	0.0%
*Total audited meat products*	*195*		*196*		*29*	

FAD is fish aggregating device, MSC is Marine Stewardship Council.

**Table 5 nutrients-11-00853-t005:** Sustainability statements and claims present on Australian supermarket own brand foods: ethical sourcing.

Food Group	Coles			Woolworths		
	Frequency	Percent	Total Audited Products	Frequency	Percent	Total Audited Products
Chocolate	11	57.9%	19	3	37.5%	8
Cooking chocolate	4	44.4%	9	0	0.0%	8
Hot chocolate	1	50.0%	2	0	0.0%	0
Sugar and syrups	7	77.8%	9	9	81.8%	11
Coffee	5	50.0%	10	18	69.2%	26
Tea	3	75.0%	4	0	0.0%	1
Total	31	58.5%	53	30	55.6%	54

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
