# Peer review of "The Nature and Quality of Australian Supermarkets’ Policies That Can Impact Public Health Nutrition, and Evidence of Their Practical Application: A Cross-Sectional Study"

_nutrients, 2019, doi:10.3390/nu11040853_

Round 1

Reviewer 1 Report

This is a clearly written and very well executed piece of research.  The authors address a very key aspect in therms of retailer role/responsibility in the role of healthy diets for consumers.  I recommend this paper for publication subject to addressing  the following minor comments.  

Line 47 reference number 9.  They are citing one paper (albeit a review) declaring that supermarkets encourage consumption of nutrient poor foods.  If making such a declaration more than one reference should be cited and acknowledge if contrary findings have been published.

Section 1.3 Doesn't address the aspect of lower carbon footprint and sustainable processing of foods resulting in a lower footprint.  This is a very large aspect of sustainable foods systems.  Please give reason as to why it was omitted or alternatively include some mention of it.

Lines 402-403.  making reduction publicly available does not always work and for certain products the producers take a stealth approach, that is, if the consumer does not know that the product is reduced salt for example, then they will not perceive an inferior taste.  In many instances there is better consumer compliance and consumption when the stealth approach is taken.  There is a huge body of evidence in this area of behavioural economics showing the effectiveness of stealth and nudge theories.

Author Response

Response to reviewer 1 comments and suggestions for authors

Thank you for reviewing and providing comments on our study.  Specific responses are provided below, and where changes have been made in the revised manuscript details are provided.  All changes have been marked in the manuscript using tracked changes. 

This is a clearly written and very well executed piece of research. The authors address a very key aspect in terms of retailer role/responsibility in the role of healthy diets for consumers. I recommend this paper for publication subject to addressing the following minor comments.

Thank you for your positive feedback on our paper.

Line 47 reference number 9. They are citing one paper (albeit a review) declaring that supermarkets encourage consumption of nutrient poor foods. If making such a declaration more than one reference should be cited and acknowledge if contrary findings have been published.

The wording has been modified, and two more references added to emphasise the impact of global supermarket development on population diets:

“Globally, there has been rapid growth in the proportion of foods sold from supermarkets, impacting population diets by making nutrient-poor processed foods more widely available [9][1, 2].” (Lines 45-48)

Section 1.3 doesn't address the aspect of lower carbon footprint and sustainable processing of foods resulting in a lower footprint. This is a very large aspect of sustainable foods systems. Please give reason as to why it was omitted or alternatively include some mention of it.

Section 1.3 describes the global initiatives led by the United Nations and the World Health Organization that call for food companies including supermarkets to take action to improve poor diets and promote sustainable food systems.  Whilst the authors agree that reducing the carbon footprint of processed food is important, the global initiatives do not specifically refer to the carbon footprint of food, and do not recommend actions that could be taken to lower the carbon footprint of processed food. 

Lines 402-403. Making reduction publicly available does not always work and for certain products the producers take a stealth approach, that is, if the consumer does not know that the product is reduced salt for example, then they will not perceive an inferior taste. In many instances there is better consumer compliance and consumption when the stealth approach is taken.  There is a huge body of evidence in this area of behavioural economics showing the effectiveness of stealth and nudge theories.

This section refers to the need for supermarkets to provide nutrient reduction targets in their corporate social responsibility reports, to improve transparency and credibility for their initiatives.  We do not argue that the information should be made available to consumers, as we agree that stealth reformulation can be more readily accepted.  

Reviewer 2 Report

Dear authors, this is a very well written paper and interessting study, it is of high interest for public health issues. It is - by my judging - the first known study concerning nature and quality of (Australian) supermarkets in respect to CSR.

General remarks:

You refer to the term "audit" which is often used in another context such as "certification or labelling…". Please describe in more detail that you actually refer to "...observe, collect, judge, evaluate…"

Please also refer in the introduction  to global/EU recommendations, regulations, implementations on CSR  and in which way CSR are different from sustainability reports...

line 212 do you mean SOBF instead of "SOB"?

how do you define and discriminate the terms "accessibility, availability, affordability"? (lines 287ff)

lines 323-331 why is it in bold letters?

line 324 there is no table 4! therefore you have then table 5 line 332....,

line 337 please explain CSR practice

lines 484 conclusions

I wonder if you could  give some general practical advices how to develop a sort of "best practice" approach for CSR policies and evaluations of it in supermarkets for practcal applications in other countries??

Methods:

There are methodological issues which I feel need some more clarifications and detail:

identification of SOBF (lines 208 ff):

how did you select those SOBF and could describe what exactly you did to to identify the products (via app?, scan data, data base) (lines 218 ff).

line234 what do you mean when you refer to "deductive thematic content analysis"?

line 238 what is a "political CRS lens"?

could you in general point to the issues of "policies" versus "practices" what do you mean by that? I think this is important for the conclusions...

Author Response

Response to reviewer 2 comments and suggestions for authors

Thank you for reviewing and providing comments on our study.  Specific responses are provided below, and where changes have been made in the revised manuscript details are provided.  All changes have been marked in the manuscript using tracked changes. 

Dear authors, this is a very well written paper and interesting study, it is of high interest for public health issues. It is - by my judging - the first known study concerning nature and quality of (Australian) supermarkets in respect to CSR.

Thank you for your positive feedback on our paper.

General remarks:

You refer to the term "audit" which is often used in another context such as "certification or labelling…” Please describe in more detail that you actually refer to "...observe, collect, judge, evaluate…"

The supermarket audits involved collecting observation information in a systematic way from each of the included stores.  For clarity, we have prefixed audit with ‘observational’ on the following lines: 26, 190 and 210.  We have also described the audits in more detail to ensure readers understand that we did not undertake enforcement audits (e.g. of food safety standards or food labelling).

“The supermarket audit methods involved collecting observational data in a systematic way from each of the selected supermarkets.  The protocol for the audits is provided in detail elsewhere and described briefly below [78]. (Lines 216-218)

Please also refer in the introduction to global/EU recommendations, regulations, implementations on CSR and in which way CSR are different from sustainability reports...

The following text has been added to section 1.3 in the introduction.

Another influence over supermarket policy-making is the requirement for large companies operating within the European Union to disclose non-financial information relating to the environment, society and governance [3].  Companies are encouraged to provide forward-looking benchmarks and commitments, and unbiased reports which balance positive and negative aspects [3].  The focus on environmental impact but not public health is consistent with the focus of other global initiatives such as the Global Reporting Initiative [4], FTSE4Good [5], the Dow Jones Sustainability index [6], and the UN Global Compact [7].” (Lines 124-130)

Line 212 do you mean SOBF instead of "SOB"?

The list provided is of the supermarket own brands identified in the observational audits, rather than a list of all the individual foods carrying supermarket own brands on packaging.  Therefore, no changes have been made to the text (line 225).

How do you define and discriminate the terms "accessibility, availability, affordability"? (lines 287ff)

Definitions of what is meant by the terms accessibility, availability and affordability in relation to this study is provided in the first column of Table 1 (Line 297).  This column provides definitions or examples for all of the attributes of public health nutrition included in the analysis.

Accessibility (i.e. location of stores, location of products, education and promotion initiatives to support selection of healthy foods)

Availability (i.e. availability of foods to meet specific needs, including healthy and sustainable)

Food cost and affordability (i.e. makes healthy foods affordable)

Lines 323-331 why is it in bold letters?

The text is not meant to be in bold letters, so it has been changed to standard text in the revised manuscript (lines 339-347).

Line 324 there is no table 4! therefore you have then table 5 line 332....,

Apologies for the omission of table 4, it has been added to the revised manuscript (line 348).

Line 337 please explain CSR practice

Wording has been changed from ‘CSR practice’ to ‘practical application of CSR’ (line 366). 

The wording of section 4.5 has also been modified, as follows:

Practical application of CSR (i.e. actions that have been implemented) has a more direct influence on food environments than CSR policies (i.e. a statement of intent), so including product information in monitoring is recommended [99].” (Lines 464-466)

Lines 484 conclusions: I wonder if you could give some general practical advices how to develop a sort of "best practice" approach for CSR policies and evaluations of it in supermarkets for practical applications in other countries??

The following text has been added to the conclusion:

Recommendations for supermarket CSR policies in Australia, and other countries, include: provide specific details, set transparent targets, and report progress against these targets; set targets for the availability, accessibility, and affordability of nutritious SOBF; reinforce the importance of food safety and quality by making existing standards more transparent; specify nutrient reduction targets, and report progress made for SOBF reformulation; ensure sustainable sourcing policies encompass all SOBF, with a focus on supporting healthy and sustainable diets.” (Lines 526-531)

Methods:

There are methodological issues which I feel need some more clarifications and detail:

Identification of SOBF (lines 208 ff):

How did you select those SOBF and could describe what exactly you did to identify the products (via app?, scan data, data base) (lines 218 ff).

Section 2.3.3 describes the data collection process, whereby two researchers visited each of the selected stores together to conduct photographic audits of all SOBF present over a 3-week period.  We did not use an app, or any other data source, so that the study provided a ‘moment-in-time’ examination of the in-store environment.  To assist with identifying the SOBF during the audits, the online shopping websites of two retailers were searched for information about their own brands, and a list was generated.  The following text has been added to clarify:

“SOBF were identified by presence of supermarket branding on the front-of-pack.  Online shopping websites for Coles and Woolworths were searched to generate lists of SOBF, which assisted with identifying the SOBF during the audits [80, 81].” (Lines 220-222)

Line234 what do you mean when you refer to "deductive thematic content analysis"?

Our deductive thematic content analysis used a predefined list of themes based on prior research, and then identified content that was consistent with each of the themes in the research materials.  We have simplified this in the manuscript as follows:

Content analysis of the CSR research materials was conducted by applying a framework of supermarket impacts on public health…” (Lines 248-249)

Line 238 what is a "political CRS lens"?

A definition for political CSR theories is provided on lines 137-139.  This has been added to section 2.4 as follows:

“A political CSR lens (i.e. whereby large, powerful companies need to act as good corporate citizens, taking responsibility for impacting society to protect their position and power [58]) guided the analysis of supermarket public health nutrition-related CSR policies.” (Lines 252-254)

Could you in general point to the issues of "policies" versus "practices" what do you mean by that? I think this is important for the conclusions...

The importance of evaluating practical application of CSR policies was described in section 4.5.  A few minor changes have been made to this section to clarify the difference between policies (i.e. a statement of intent) and practice (i.e. actions that are implemented in stores).

Practical application of CSR (i.e. actions that have been implemented) has a more direct influence on food environments than CSR policies (i.e. a statement of intent), so including product information in monitoring is recommended [99].  Whilst the inclusion of supermarket audits to evaluate how well CSR policies were practically applied in stores was time consuming, this study reveals several advantages including:…” (Lines 464-468)
